Comparative analysis of radiological outcomes among cephalomedullary nails: helical, screw and winged screw

Vahabi Arman armanvy@gmail.com
Dastan Ali Engin
Kilicli Bunyamin
Aljasim Omar
Gunay Huseyin huseyin.gunay@ege.edu.tr
Ozkayin Nadir
Aktuglu Kemal
Department of Orthopaedics and Traumatology, Ege University , Izmir , Turkey
Barak Meir
Electronic publication date: 2024 Sep 19
Publication date: 2024
Volume: 12
Electronic Location ID: e18020
Received 2024 Jan 25; Accepted 2024 Aug 9
Copyright: ©2024 Vahabi et al.
Copyright year: 2024
Copyright holder: Vahabi et al.
License: This is an open access article distributed under the terms of the Creative Commons Attribution License, which permits unrestricted use, distribution, reproduction and adaptation in any medium and for any purpose provided that it is properly attributed. For attribution, the original author(s), title, publication source (PeerJ) and either DOI or URL of the article must be cited.
License URL: https://creativecommons.org/licenses/by/4.0/

Keywords: Screw blade, Helical blade, Lag sliding, Neck shortening, Wedge wing, Proximal femoral nail

Funding: The authors received no funding for this work.

==============================
Objective

Cephalomedullary nails (CMN) are implants with a high success rate in the surgical treatment of trochanteric fractures. The aim of this study is to compare the radiological outcomes and mechanical complications of femoral trochanteric fractures treated with three different CMNs.

Methods

Intertrochanteric fractures in patients aged 50 years and older treated with CMN between January 2016 and December 2021 were reviewed retrospectively. A total of 158 cases meeting the criteria were included to final analysis. Cases were divided into three groups based on the type of nail used (helical blade: group 1, n = 54; screw: group 2, n = 53; winged screw: group 3, n = 51). Demographic characteristics, mechanical complications, reduction quality, tip-apex distances (TAD) and Cleveland zones were compared between the groups. Femoral neck shortening, varus collapse, lag sliding, changes in abductor length were compared between study groups. Factors affecting mechanical complications were also analyzed.

Results

Study groups were homogenic in terms of demographic characteristics, fracture type and reduction quality. Regarding mechanical complications, no statistically significant difference was found between groups. All three implants had similar outcomes on femoral neck shortening, varus collapse and lag sliding. Pooled analysis of 158 cases showed that mechanical complications increase as the quality of reduction decreases (p = 0.000) same applies when TAD alters from the desired range (p = 0.025) and with non-optimally implanted blade according to Cleveland zones (p = 0, 000).

Conclusion

The radiological outcomes and mechanical complications of helical blade, screw type blade and winged screw type blade proximal femoral nails are similar in selected group. Regardless of the device type, it is necessary to obtain high reduction quality, obtain TAD within described range and optimally place the blade according to Cleveland Zones to reduce the failure rate and avoid complications.

Introduction

Hip fractures in the elderly population are a common and growing problem world-wide. The current estimated global incidence is over 1.6 million and projections for 2050 estimates over at least 4.5 million cases (Gullberg, Johnell & Kanis, 1997; Veronese & Maggi, 2018). Considering increasing numbers both for life expectancy and population growth, optimal treatment for hip fractures would remain a major issue for healthcare providers. Especially in older population, besides direct medical expenses, loss of activity level, need for a caregiver and problems related to prolonged immobility, disability related depression are other matters of concern related to hip fractures (Roberts et al., 2015). Choosing optimal implant in such patients to avoid suboptimal results is of utmost importance for these patients. Recently, cephalomedullary nails outnumbered and replaced sliding screws, which once known as a gold standard for these types of fractures (Rogmark, Spetz & Garellick, 2010).

Optimal implant in treatment of intertrochanteric fractures is still matter of debate. Both extra and intramedullary implant types being subject to that debate, cephalomedullary nails had shown to have some advantages over extramedullary implants. Shorter operation time, better pain management, less blood loss, gradual learning curve are being some of them (Nherera et al., 2018). But with mentioned advantages, CMNs are not freed from its disadvantages. After its first clinical uses in 90s, varus collapse, cut-out, z-effect and reverse z-effects were faced complications that lead to search for better CMN type.

The CMN with a helical blade (PFNA-proximal femoral nail antirotation) was developed as an advancement over earlier PFN (proximal femoral nail) designs with two screws. Although some studies suggest that the helical blade is superior to screw-type implants in terms of complication rates (Sharma, Mahajan & John, 2017), there is still ongoing research to identify the ideal implant. The Dyna-Locking Trochanteric (DLT) Nail is a more recent PFN type that features an additional anchor-like mechanism (winged) designed to achieve better grasp of the femoral head. In this study, we focused on comparing three different types of cephalomedullary nails (helical blade, screw-type blade, and DLT type) in terms of mechanical complications, implant failures, and radiological variables such as sliding of the neck component, changes in abductor arm, vertical femoral neck shortening, and varus collapse. These parameters reflect the biomechanical properties of the implants. Our objective was to evaluate the performance of these three different implant types through radiological assessments.

Material and Method

This study conducted in Ege University School of Medicine, Department of Orthopaedics and Traumatology. Ethical approval obtained from Ege University Ethics Committee (Num: 21 1T/02). Written informed consent from every patient was obtained. Intertrochanteric fractures treated with CMN between January 2016 and December 2021 were reviewed retrospectively through electronic patient files. Patients with closed intertrochanteric hip fractures, aged 50 years or older, with at least 6 months of follow-up were included to final analysis. Cases with pathological fractures, a history of ipsilateral hip surgery, immobility prior to fracture and lack of appropriate x-rays were excluded (Fig. 1). Total of 158 patients were included in the study.

Figure 1 Patient enrollment process.

Three different types of cephalomedullary nails compared. Group A: PFNA-II helical type blade (Synthes, Oberdoff, Switzerland); group B: CMN with screw type blade (Zimmer Natural Nail, Warsaw, IN, USA); group C: Winged screw type Dyna Locking Trochanteric (DLT) Nail (U&I Corporation, Korea). AO/OTA classification system used for classifying fractures either as stable or unstable. This classification system is a commonly employed alphanumerical system for classifying fracture types in different bones. The first two numbers refer to the specific bone, and the following letter and number provide additional detail of fracture pattern. The term ‘stability’ for intertrochanteric fractures describes the resistance force of the fracture pattern to compressive loads once reduced. While 31A1 fractures classified as stable, 31A2 and 31A3 fractures classified as unstable. One of the key criteria was the reduction quality achieved for homogeneity between groups was achieved reduction quality. Quality of reduction assessed through scoring system described by Chang et al. (2015). Considering Garden alignment and fragment displacement, both in AP and lateral view; scoring 0 to 4. Quality of reductions classified as poor, acceptable and excellent.

Tip-apex distance values obtained on first postoperative X-ray through method described by Baumgaertner et al. (1995). This variable is a validated tool for assessing cut-out risk for intertrochanteric fractures treated with CMNs, as it represents the implant’s grip on the denser subchondral bone of the femoral head. TAD was measured on X-rays proportioned with known diameter of the neck screw/helical blade. Distance between the tip of the screw and apex of femoral head measured in both AP and lateral hip X-rays. Values between 15-25 mm considered appropriate (van Leur et al., 2019).

Values for lag screw sliding throughout healing process measured with technique described by Watanabe et al. (2002). Technique used described in Fig. 2. The measured values were scaled by comparing with the known diameter of the screw (Fig. 2). The term ‘lag sliding’ describes changes in the blade’s position and refers to the device’s resistance to compression along the fracture line. Extensive sliding is not intended for cephalomedullary nails (CMNs). Technique used for lag sliding measurement was as follows: First line drawn parallel to ground and intersects tip of blade. Second line drawn parallel to ground and intersects tip of nail. Distance between two lines acquired for first postop (M1) and last follow up (M2). Total sliding value calculated as T =— M1-M2— /cos50° (for nails with angle of 125, cos 55 used).

Figure 2 Technique used for lag sliding measurement described.

First line drawn parallel to ground and intersects tip of blade. Second line drawn parallel to ground and intersects tip of nail. Distance between two lines acquired for first postop (M1) and last follow up (M2). Total sliding value calculated as T = —M1-M2— /cos50° (for nails with angle of 125, cos 55 used).

To assess femoral neck shortening, modified technique adapted from Zlowodzki was used (Zlowodzki et al., 2008). To perform that, first postoperative anteroposterior xray of the hip outlined, scanned, and then superimposed on most recent follow-up xray with help of graphic software (Rhinoceros 3D, Robert McNeel & Associates, Seattle, WA, USA). Difference between first postop and last follow-up considered. Changes in vertical plane assessed as femoral vertical neck shortening and changes in horizontal plane assessed as abductor arm shortening. Scaling done by comparing with known diameter of blade.

Changes in neck-shaft angle in follow-up defined as varus collapse. Varus collapse refers to the decrease in the collo-diaphyseal angle after fixation and the loss of reduction in the coronal plane. For the values of varus collapse, lag sliding and abductor arm length; Difference between first post-operative and last follow-up X-rays considered. If patients received a secondary operation due to a mechanical complication, their measured variables were not included into comparative analysis. Spatial location of the tip of blade in femoral head was recorded with Cleveland method (Cleveland et al., 1959). This classification system categorizes the position of the tip of the blade within the femoral head into nine distinct regions, represented in a 3-dimensional sphere. Cleveland zone for every hip was noted in first postoperative X-ray. Placement of blade classified as optimal, suboptimal, and moderate. Placement in Zones 2, 5 and 8 considered as optimal position. Placement of blade in Zones 4, 6, 7 and nine considered as suboptimal placements while Zones 1 and 3 were considered as moderate placement (van Leur et al., 2019).

Same postoperative weight bearing, and mobilization protocols applied to all patients. Patients allowed and encouraged weight bearing after day 1 alongside with quadriceps strengthening exercises and mechanical DVT prophylaxis. On follow-up visits scheduled at postoperative 2nd, 6th and 12th weeks. AP and lateral hip X-rays obtained in every visit. At 12th week follow-up, standing long leg radiographs obtained for accessing length discrepancy and rotational status of lower extremity. Final follow-up was decided as last follow-up X-ray.

The data obtained in the study were put into the database created in the SPSS 22 (IBM Corporation, Armonk, NY, USA) program and statistical analysis were performed with the same program. Frequencies and percentages of categorical variables, mean values, standard deviation, median values, min and max. values were calculated. The suitability of the continuous variables to the normal distribution was investigated. Non-parametric methods were preferred for the comparison of independent groups. Multiple comparisons of independent groups in continuous variables were made using Kruskal-Wallis method. Cross tables were prepared for categorical variables, distribution differences of groups were tested with chi-square method, Type 1 margin of error was determined as α:0.05 in all statistical comparisons. Level of significance was assigned as p < 0.05.

Results

Mean age was 73.8 (± 10.4) years ranging from 50 to 96. In group 1 (Helical) mean age of 73.1 (±10.4) years ranging from 50 to 89 years. In group 2 (Screw) mean age of 74.1 (±10.2) years ranging from 50 to 96 years. In group 3 (winged screw) mean age of 74.3 (±10.9) years ranging from 52 to 94 years. (p = 0.848). Gender distribution in our series was 60 male and 98 female patients. Group I (15 M, 39 F), group 2 (20 M, 33 F), group 3 (25 M, 26 F) (p = 0.081). Distribution between groups was homogeneous in terms of stability of fracture according to AO/OTA classification (p = 0.196). Mean follow-up period of groups 1, 2 and 3 were 19.02 ± 14.04, 18.14 ± 12.94 and 16.19 ± 8.79 respectively. There were no statistically significant differences in terms of follow-up duration among the study groups (p = 0.568) (Table 1).

Distribution of reduction qualities as excellent or acceptable/poor were similar in all three groups (p = 0.855) (Table 2). Distribution of optimally implanted blade according to Cleveland index (p = 0.252) were homogenous also.

Reduction quality was strongly related with complication rates (p = 0.000). Tip-apex distance values were similar in all groups (p = 0.443). TAD values classified in two groups as appropriate or inappropriate. Appropriate TAD values were associated with lower complication rate while inappropriate values of TAD were associated with higher complication rates (p = 0.025). Moderate and suboptimaly implanted blades (according to Cleveland Zones) were significantly related with mechanical complications (p = 0.000).

Total of 29 patients faced mechanical complications (Table 3). Cut-out was most common (n = 17, 10.7%) (Fig. 3) mechanical complication that was followed by peri-implant fracture (n = 4, 2.5%) (Fig. 4). Other complications were non-union (n = 4, 2.5%) (Fig. 5) implant failure (n = 4, 1.2%) (Fig. 6). and avascular necrosis (n = 1, 0.6%). Comparing mechanical complication rates, no statistically significant difference was found between three groups. Pooled analysis of 158 cases, showed that mechanical complications increased as the reduction quality decreased (p = 0,000), when TAD is not in the desired range (p = 0.025) and when blade is not implanted optimally according to Cleveland index (p = 0,000).

Table 1 Demographic characteristics and fracture patterns.

		Group 1 (Helical)	Group 2 (Screw)	Group 3 (DLT)	p value	
Gender: n (%)	Male	15 (27.8%)	20 (37.7%)	25 (49%)	0.081	
Female	39 (72.2%)	33 (62.3%)	26 (51%)	
Age (Mean ± SD)		73.1 ± 10.4	74.1 ± 10.2	74.3 ± 10.9	0.848	
Fracture Stability n (%)	Stabile	22 (40.7%)	29 (54.7%)	29 (56.9%)	0.196	
Unstable	32 (59.3%)	24 (45.3%)	22 (43.1%)	
Follow-up (Mean ± SD)		19.1 ± 14.1	16.83 ± 12.4	15.48 ± 8.67	0.568	
Notes.

SD: Standard Deviation

Table 2 Comparison of reduction quality, tip-apex distance and Cleveland index between groups.

		Groups		
		Group 1
n (%)	Group 2
n (%)	Group 3
n (%)	p value	
Reduction quality n (%)	Excellent	33 (61.1%)	34 (64.2%)	30 (58.8%)	0.855	
Acceptable and poor	21 (38.9%)	19 (35.8%)	21 (41.2%)	
Tip-apex distance n (%)	Appropriate	20 (37.0%)	26 (49.1%)	21 (41.2%)	0.443	
Inappropriate	34 (63.0%)	27 (50.9%)	30 (58.8%)	
Cleveland index n (%)	Optimal	20 (37.0%)	23 (43.4%)	16 (31.4%)	0.252	
Suboptimal	19 (35.2%)	19 (35.8%)	27 (52.9%)	
Moderate	15 (27.8%)	11 (20.8%)	8 (15.7%)	

Table 3 Number of complications and distribution of complications between groups.

	Group 1
(n = 54)	Group 2
(n = 53)	Group 3
(n = 51)	Total
(n = 158)	
Cut-Out	5	7	5	17	
Non-union	2	1	1	4	
Avascular necrosis	1	0	0	1	
Implant failure	1	1	0	2	
Peri implant fracture	4	0	0	4	
Complications (total)	13	9	7	29	

Figure 3 Cases of cut-out.

(A) Cut-out case in group 1. (B) Cut-out case in group 2. (C) Cut-out case in group 3.

Figure 4 Case of peri-implant fracture below the implant.

Figure 5 Cases of non-union.

(A) Non-union case in group 1. (B) Non-union case in group 2. (C) Non-union case in group 3.

Figure 6 Cases of implant failure.

(A) Implant failure case in group 1. (B) Implant failure case in group 2.

Complication rate was 18.4%. In group 1 (Helical) it was 24.1%, in group 2 (Screw) it was 17.0% and in group 3 (DLT) it was 13.7%. Difference of complication rate between study groups were statistically insignificant. Total of four peri-implant fractures were observed in our series and all four of them were in group A but there this variable was not statistically significant.

Evaluating varus collapse, there was no statistically significance difference between groups (p = 0.954). All three implants managed to prevent varus collapse effectively. For the vertical shortening of femoral head measurements there were no significant differences while there were slightly less shortening in group A (2.00 vs 3.35 and 4.80 respectively) (p = 0.343). Horizontal neck shortening which can be interpreted as abductor arm shortening were similar in all groups while in group C there were less shortening (0.35 vs 2.10 and 3.30) (p = 0.115). Lag sliding measurement were similar between all three groups while in group C median values were higher, but this was not statistically significant (Table 4).

Table 4 Radiological outcomes in patients without complications.

		Num	Median	Percentile 25	Percentile 75	p value	
Varus collapse
(In degree)	Helical	41	3.00	0.00	5.00	0.954	
Screw	44	3.00	0.00	5.00	
DLT	44	3.00	0.00	6.50	
Vertical femoral neck shortening
(In millimeters)	Helical	41	2.00	0.00	5.40	0.343	
Screw	44	3.35	0.00	7.35	
DLT	44	4.80	0.65	8.40	
Horizontal femoral neck shortening
(In millimeters)	Helical	41	2.10	0.00	5.20	0.115	
Screw	44	3.30	0.00	7.25	
DLT	44	0.35	0.00	3.75	
Lag sliding	Helical	41	0.00	0.00	1.24	0.063	
Screw	44	0.21	0.00	4.20	
DLT	44	1.04	0.00	4.98	

Regarding subgroup analysis based on fracture stability, complication rates did not significantly differ among the three implant types when unstable fractures were evaluated individually (p = 0.311). For stable fractures, the distribution of complications was insufficient for statistical analysis. In the subgroup analysis of radiological variables for stable fractures, no significant differences were observed for varus collapse, vertical shortening, horizontal shortening, and lag sliding distance (p = 0.250, p = 0.712, p = 0.850, and p = 0.512, respectively). Similarly, when these parameters were assessed in unstable fractures, there were no significant differences between the groups for varus collapse, vertical shortening, and horizontal shortening (p = 0.418, p = 0.072, and p = 0.092, respectively). However, a significant difference in lag sliding distance was noted between group 3 and group 2, favoring group 2 (p = 0.015).

Discussion

Helical blade (PFNA), screw-type blade (ZNN), and winged screw (DLT) nails were all effective in the treatment of intertrochanteric fractures in patients aged over 50. These devices demonstrated similar success in preventing varus collapse, lag sliding distance, vertical femoral neck shortening, horizontal femoral neck shortening, and mechanical complication rates.

The limitations of this study included its retrospective nature and the limited number of patients due to a high rate of loss to follow-up. The relatively short mean follow-up period could also be considered a limitation. However, given that most mechanical complications occur within the first three months of follow-up, the mean value of 12.7 months can be deemed acceptable for the purposes of this study. Another acknowledged shortcoming was the lack of homogeneity testing of bone mineral density between groups.

Hip fractures in the elderly population present numerous challenges. Factors such as comorbidities, bone quality, fracture stability, and pre-injury activity levels are beyond our control. However, selecting the optimal implant, achieving good reduction quality and appropriate TAD (tip-apex distance), and implementing suitable postoperative rehabilitation programs are within our control. Additionally, institutional and local facilities, as well as variable access to implants, may influence implant selection beyond the surgeon’s preference. Within the scope of the variables evaluated in this study, our results did not indicate any mechanical or radiological superiority to highlight one device over another.

Using intramedullary nailing techniques, fractures heal through secondary bone healing. In these techniques, gains achieved by the minimally invasive surgeries are countered by relatively bigger gap at the fracture line and lesser stability of fracture than absolute fixation (Norris et al., 2018). Intraoperative compression acquired through implants aim to minimize this risk by reducing the fracture gap. Too big of a gap or continuous motion through sliding on a fracture line could disrupt bone healing. Regarding individualistic nature of healing, defining “healing” on a scale that could enclose all variables is a challenging task. However, reoperation rate suggested as a good tool for this purpose (Claes, 2021). Our results, comparing mentioned device types, are suggesting no difference in reoperation rate between groups. This can be interpreted as all three devices were similar regarding final healing status and healing rate.

Term “lag sliding” in a CMN defines amount of sliding of a blade/screw component through nail. These types of implants are designed to allow compression on fracture site through sliding of blade on nail with a fixed angle, to achieve fracture healing and load sharing. Fail to do so could end up with restricted healing, implant failures or blade protrusion through acetabulum (Takemoto et al., 2014). Excessive sliding could cause irritation on soft tissues that results with lateral pain, or even may result with non-union due to instability at fracture line (Lim et al., 2021). Difference in distribution of lag sliding values between groups could have mask other structural variables between groups but that was not the case in our series. Homogeneous distribution of lag sliding values in all three groups suggested this effect of implant is negligible while evaluating other possible factors for mechanical complications.

Excessive femoral neck shortening after pertrochanteric femur fractures could be a functionally impairing complication. On the other hand, there is evidence for benefits of sliding and compression for fracture healing. But how much compression should we apply intraoperatively? In expense of shortening femoral neck, what extend of compression obtained through sliding remains beneficiary? Current literature fails to suggest level 1 evidence for these questions. But based on our current knowledge, we can conclude that optimal implant should ensure compression at fracture site intraoperatively while maintaining femoral neck length through healing process. Excessive compression could ease healing while disrupting neck length, abductor balance and gait dynamics. Although disrupted abductor balance mainly effects clinical outcomes, in our study, we investigated changes in abductor arm length as an indicator of device’s mechanical success to preserve initial length. However, clinical manifestation of changes in abductor arm length in hip procedures is not revealed in all aspects. While some reports suggested that increasing femoral offset is associated with better functional outcomes in arthroplasty procedures (Clement et al., 2016), some others reported better outcomes with shortened femoral offset after trochanteric fractures (Buecking et al., 2015).

Achieving the desired TAD, which means that the ends of the screws are placed in the denser subchondral bone, provides higher resistance to cut-out. Our results for link between cut-out and TAD was compatible with described ranges. A review of 10 studies, published by Li et al. (2015) reported lower cut-out rates with helical blade compared to lag screw. Another review in 2021 with data from 21 studies suggesting otherwise, reported no difference between two implant types (Ng et al., 2022). In our study, there were no difference between groups in that manner. Meta-analysis on these comparisons mostly failed to infer a solid data with complications other than cut-out. Lately published meta-analysis by Kim et al. (2021) suggested even higher fixation failures in helical group, especially cut-through. A study by Kim et al. (2013) focused on the comparison of cut-out rates of DLT, Gamma 3, and PFNA suggested no difference between groups in terms of studied radiological variables and clinical outcomes.

The alleged superiority of the helical blade over screw type blades is based on the theory that it can be inserted without drilling which causes further loss of precious bone stock in femoral head, thus it provides a better anchoring with the help of bone impaction during insertion of blade. Several biomechanical studies were focused on the comparison of helical and screw blade. Study by Kwak et al. compared screw type (Gamma 3), hybrid type (Gamma 3 U blade) and blade type (PFNA-II) and reported better rotational stability of the femoral head in blade and hybrid group. However, resistance to varus collapse is founded to be greater in hybrid blade group, compared to helical blade and migration of screw (/blade) in femoral head was greater in blade group (Kwak et al., 2018). Finite element analysis by Chen et al. (2020) comparing PFN-II and ZNN on a reverse oblique intertrochanteric fracture model reported similar displacements but reduced stress on both bone and implant in ZNN group. Our study’s results were compatible, suggesting no differences between devices. To best of our knowledge, there is no published article on the biomechanics of the DLT nail. Well-designed biomechanical studies can provide more objective data on comparison between these three devices.

A randomized controlled trial from Korea by Shin et al. (2017) with 353 patients, compared ZNN and PFNA. Primary objective was to compare functional outcomes, but reoperation rate and cut-out incidence were also analyzed. While clinical comparison failed to suggest any significant difference between two devices, data on complications was compatible with our study, suggesting implant type alone is not a significant variable for cut-out. Univariate analysis suggested that TAD was the major predictor for cut-out. Reduction quality was not related to cut-out rate (Shin et al., 2017). There were limited number of published studies about winged screw type DLT nail in published literature, and few of them were designed as a comparative study. Retrospectively conducted case series by Temiz, Durak & Atici (2015), Baik et al. (2021) and Gunay et al. (2014) reported satisfactory clinical and radiologic results with limitations of their designs: they all were single center studies with limited number of patients. To best our knowledge this study is first to compare winged screw design device to other commonly used cephalomedullary nails in settings of mechanical complications. Possible future reviews with more reported series could suggest higher level data.

Conclusion

Helical blade, screw and winged-screw type nails were all effective in treatment for intertrochanteric fractures in patients aged over 50 with similar mechanical complication rates. Search for optimal implant continues and so far, modified implants with good biomechanical study results fail to show same success in clinical aspect. Radiological parameters related to implant type (neck shortening, lag sliding and varus collapse) showed no significant differences between the compared implant types. Obtaining good reduction and achieving described TAD values are key factors to prevent mechanical complications.

Supplemental Information

Supplemental Information 1 STROBE Checklist

Additional Information and Declarations

Competing Interests

Author Contributions

Human Ethics

Data Availability

The authors declare there are no competing interests.

Arman Vahabi conceived and designed the experiments, performed the experiments, analyzed the data, prepared figures and/or tables, authored or reviewed drafts of the article, and approved the final draft.

Ali Engin Dastan conceived and designed the experiments, analyzed the data, prepared figures and/or tables, and approved the final draft.

Bunyamin Kilicli performed the experiments, prepared figures and/or tables, and approved the final draft.

Omar Aljasim conceived and designed the experiments, performed the experiments, authored or reviewed drafts of the article, and approved the final draft.

Huseyin Gunay conceived and designed the experiments, authored or reviewed drafts of the article, and approved the final draft.

Nadir Ozkayin conceived and designed the experiments, authored or reviewed drafts of the article, and approved the final draft.

Kemal Aktuglu conceived and designed the experiments, authored or reviewed drafts of the article, and approved the final draft.

The following information was supplied relating to ethical approvals (i.e., approving body and any reference numbers):

This study was conducted in Ege University School of Medicine, Department of Orthopedics. Ethical approval obtained from Ethics Committee of Ege University (Num: 21 1T/02).

The following information was supplied regarding data availability:

The dataset for complete analyzed measurements are available in the Supplemental File.

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
