# Peer review of "Comparative analysis of radiological outcomes among cephalomedullary nails: helical, screw and winged screw"

_PeerJ, doi:10.7717/peerj.18020_

## Round 0.1 · original submission · Major Revisions

Dear Drs. Vahabi and Günay,

Your manuscript titled " Comparative analysis of radiological outcomes among cephalomedullary nails: PFNA, ZNN, and DLT" was considered by three expert reviewers and based on their opinions and my review, the decision is “Major revisions”.

Please carefully read the reviewers’ comments and address them fully in your revised manuscript. In addition, please address the following points:

(1) Manuscript and supplementary table (excel) must be fully in English. Please review all text and correct (e.g., table 3 is titled “Tablo”, many entries in the supplementary table are not in English).

(2) The supplementary table (excel) lists 161 patients while the manuscript states there were only 158 patients (L79, Figure 1). Please explain the discrepancy.

(3) The study focuses on “Radiological outcomes” – why are they important? Why not look at the clinical outcomes? Please explain

(4) Please spell out the full abbreviation the first time you use it (e.g., abstract L15 DLT, intro L59 PFN, etc.).

(5) PeerJ is a general science journal, not solely aimed at the orthopedic community. Thus, all terms and jargon must be explained. In many cases the authors use orthopedics terminology which will not be understood by the typical PeerJ reader. Here are several examples (not an exhaustive list):
L56-7: “varus collapse, cut-out, z effect and reverse z-effects” all terms need to be explained. L85: “While 31A1 fractures classified as stable, 31A2 and 31A3 fractures classified as unstable”. Paragraphs in L100-106 and 108-116. Here, figures will help (as indicated by one of the reviewers).
L112-3: “Cleveland method” and zones. The typical reader of PeerJ would have no idea what these are.
L144 “AO/OTA classification”.
L257 “RCT”.
Please add short explanation to the manuscript whenever a technical term is used. Figures would be helpful too (as asked by one of the reviewers).

(6) Decimal point symbol must be a period, not a comma (see results section and tables).

(7) Consistency – the 3 groups are called differently at different parts of the manuscript: 1/2/3, I/II/III. Please choose one and stay with it throughout the manuscript.

(8) Tables/figure legends are very short and missing info.

(9) “thru” is a nonstandard spelling of “through”. Please correct throughout the manuscript.

(10) L252: “Until December 2022, the time this study was submitted…”. This is a sentence from a previous version/submission. More than 18 months have passed since then. The authors need to research the literature for any new studies and update their findings (or update the date).

(11) Figure 2: please indicate on the figure what is the blade and what is the nail (again, PeerJ is not a surgery specific journal), color coding could also help. In addition, the 2 images in the figure look identical (i.e., M1 = M2). Please modify M@ to better explain “lag sliding”.

(12) Figure 2: “Technique used for lag sliding measurement described. First line drawn parallel to ground and intersects tip of blade. Second line drawn parallel to ground and intersects tip of nail. Distance between two lines acquired for first postop (M1) and last follow up (M2). Total Sliding value Calculated as T=| M1-M2| /cos50° (for nails with angle of 125, cos 55 used)”. This info (legend) must appear in the materials and method.

(13) Table 3: add a final total column, as this total number is referred to in the results section.

Please ensure that all review, editorial, and staff comments are addressed in a response letter and any edits or clarifications mentioned in the letter are also inserted into the revised manuscript where appropriate.

Please note that submitting a revision of your manuscript does not guarantee eventual acceptance, and that your revision may be subject to re-review by the reviewer(s) before a decision is rendered.

Reviewer 1 ·

Basic reporting

Address of corresponding author required before abstract

A better grasp of English is required at the following lines
Line 61 – a long way to go
Line 62 – one of the later
Line 63 – in this study
Line 75,87,196,197,205,218– through
Line 85,86 – one of the key criteria wad the reduction quality achieved
Line 93 – apex of the femoral head
Line 100 – shortening, modified technique
Line 104,108– f/u xray
Line 106 – ‘Thru' to be replaced with 'by'
Line 109 – arm length, difference
Line 111 – was not included
Line 119 – alongside ( cancel with )
Line 120 – on
Line 126 – analysis
Line 139-179 All figures in decimal points (not in commas)
Line 161 – that was followed by
Line 162 – No fullstop after implant failure
Line 221 – while maintaining throughout the

Experimental design

Comparison needed between the outcome in stable and unstable fractures between the 3 implants

Definition for excellent, acceptable and poor reduction quality?

Kindly mention bone quality of these patients as well as osteoporosis has an adverse impact on the outcome

Validity of the findings

This study deals only with radiological parametres without clinical correlation. A clinical correlation would have better identified the optimal implant to be used.
Difficult to say that all 3 implants fare the same just by radiological parametres. Kindly provide clinical correlation if available

Reviewer 2 ·

Basic reporting

Dear Editor and authors,
I am honored that you sent me this article for review. As an orthopedic trauma surgeon, I read your article with great curiosity and interest.
The manuscript is well-written but lacks images of important complications:
1- please add the images of the cutout cases, one from each group side-by-side.
2- please add the images of the nonunion cases, one from each group side-by-side.
3- please add the image of the only AVN case.
4- please add the images of the two implant failure cases side-by-side.
5- please add an image of the peri-implant fracture.

Let me see the revised version thereafter.
Kind regards.

Experimental design

.

Validity of the findings

.

Reviewer 3 ·

Basic reporting

It is recommended to give the explanation of abbreviations under the table. For example SD ? Indicate that the numbers given in Table 3 are numbers and give the abbreviation explanation under the table.

Experimental design

- Why were hip fractures under 50 years of age exclusion criteria?
-''All 3 implants managed to prevent varus collapse effectively.'' It is recommended to explain in detail that varus collapse is a condition related to the implant type or to remove this sentence.
-Discussion paragraph must begin with a concise results explanation, follow by limits of the study, and the proper discussion section after these, according to the STROBE guidelines that you can follow to ensure a higher scientific soundness of your work.

Validity of the findings

-‘’ There was no study focused on the comparison of cut-out rates of DLT and other CMN types at the time this study was conducted.’’ This sentence needs to be revised. There are studies reporting high rates of cut-out and implant failure with winged nails. In addition, it is an interesting finding that cut-out rates were lower in the DLT group in your study.

-It is already known that the quality of reduction in hip fractures is correlated with functional results. What is the contribution of your study to the literature?

Additional comments

Dear Author, Thank you for the opportunity to review the article entitled "Comparative Analysis of Radiological Outcomes among Cephalomedullary Nails: PFNA, ZNN, and DLT'' thank you for giving me the opportunity to review the manuscript. This manuscript about hip fractures, which is one of the important problems seen in geriatrics, may contribute to the literature. Nevertheless, some corrections are needed in the manuscript.

---

## Round 0.2 · Minor Revisions

Dear Drs. Vahabi and Günay,

Your manuscript titled " Comparative analysis of radiological outcomes among cephalomedullary nails: Helical, screw and winged screw" was reconsidered by two expert reviewers and based on their opinions and my review, the decision is “Minor revisions”.

Please address the following points:
(1) Raw data (excel file) is still not fully in English (see note column for example). Please go over the entire supplementary table and make sure it is all in English.
(2) I can see now that there are only 158 patients. However, the numbering still implies that 4 patients (i.e., #2, 8, 117, 133) from the original cohort were removed. Please explain. It would help if this explanation will also be in the manuscript.

Please ensure that all review, editorial, and staff comments are addressed in a response letter and any edits or clarifications mentioned in the letter are also inserted into the revised manuscript where appropriate.

Please note that submitting a revision of your manuscript does not guarantee eventual acceptance, and that your revision may be subject to re-review by the reviewer(s) before a decision is rendered.

Reviewer 2 ·

Basic reporting

Thank you for the hard work and exciting case figures. I recommend the acceptance of this article.

Experimental design

Thank you for the hard work and exciting case figures. I recommend the acceptance of this article.

Validity of the findings

Thank you for the hard work and exciting case figures. I recommend the acceptance of this article.

Reviewer 3 ·

Basic reporting

no comment

Experimental design

no comment

Validity of the findings

no comment

Additional comments

Proposed edits were made by the authors.May be published.Thank you for the opportunity to review this manuscript.

---

## Round 0.3 · accepted · Accept

Dear Drs. Vahabi and Günay,

Thank you for submitting your revised manuscript titled " Comparative analysis of radiological outcomes among cephalomedullary nails: Helical, screw and winged screw". After reading the revised manuscript I’m happy to let you know that decision is “accept”.